# Anti-Metatype Antibody Screening, Sandwich Immunoassay Development, and Structural Insights for β-Lactams Based on Penicillin Binding Protein

**DOI:** 10.3390/molecules26185569

**Published:** 2021-09-13

**Authors:** Yuchen Bai, Leina Dou, Weilin Wu, Zhimin Lu, Jiaqian Kou, Jianzhong Shen, Kai Wen, Zhanhui Wang

**Affiliations:** College of Veterinary Medicine, China Agricultural University, Beijing Key Laboratory of Detection Technology for Animal Derived Food Safety, Beijing Laboratory for Food Quality and Safety, Beijing 100193, China; BS20193050465@cau.edu.cn (Y.B.); echodou17@163.com (L.D.); 2016305010312@cau.edu.cn (W.W.); S20203050782@cau.edu.cn (Z.L.); kjq2182266753@163.com (J.K.); sjz@cau.edu.cn (J.S.); wangzhanhui@cau.edu.cn (Z.W.)

**Keywords:** sandwich immunoassay, β-lactams, PBP2x, anti-metatype antibody, computational chemistry, molecular recognition

## Abstract

Theoretically, sandwich immunoassay is more sensitive and has a wider working range than that of competitive format. However, it has been thought that small molecules cannot be detected by the sandwich format due to their limited size. In the present study, we proposed a novel strategy for achieving sandwich immunoassay of β-lactams with low molecular weights. Firstly, five β-lactam antibiotics were selected to bind with penicillin binding protein (PBP)2x* to form complexes. Then, monoclonal and polyclonal antibodies against PBP2x*-β-lactams complexes were produced by animal immunization. Subsequently, the optimal pairing antibodies were utilized to establish sandwich immunoassay for detection of 18 PBP2x*-β-lactam complexes. Among them, ceftriaxone could be detected at as low as 1.65 ng/mL with working range of 1–1000 ng/mL in milk. To reveal the detection mechanism, computational chemistry and molecular recognition study were carried out. The results showed that β-lactams with a large size and complex structures maybe conducive to induce conformational changes of PBP2x*, and then exhibit greater possibility of being detected by sandwich immunoassay after combination with PBP2x*. This study provides insights for subsequent investigations of anti-metatype antibody screening and sandwich immunoassay establishment for small-molecule detection.

## 1. Introduction

Immunoassays are nowadays being employed for rapid contaminant analysis in clinical, environmental, and food samples [1,2]. The formats of immunoassay are roughly categorized as either non-competitive or competitive. The competitive immunoassay is based on the competition between protein-labeled analyte and the analyte for limited antibody amount. Conversely, the non-competitive immunoassay, generally known as the sandwich format, involves the use of two antibodies (one for capturing analyte and another labeled by an enzyme for detection), which can be added in excess [3,4]. Typically, the sandwich immunoassay yields considerable benefits in sensitivity, working range, and specificity compared to the competitive format. However, small molecules with molecular weights below 1000 Da are generally detected based on the competitive format due to the steric hindrance [5]. Many attempts have been made to establish non-competitive assays for small molecules, including the use of anti-idiotypic antibodies, anti-metatype antibodies, and recombinant VH and VL to form an open sandwich immunoassay [6,7,8,9,10]. Among these, the use of anti-metatype antibodies closely resembles the sandwich immunoassay. The anti-metatype antibodies are developed from animals immunized with the receptor-ligand complex, which may recognize parts of the structure of both the ligand and receptor at the same time or the conformational change on the receptor induced by binding, but exhibit very low or no affinity for the receptor or the ligand alone [5]. The success of the development of anti-metatype antibodies depends on the stability of the receptor–ligand complex [9].

β-lactams refer to a class of antibiotics with a β-lactam ring in their chemical structure, which destroys the cell wall of bacteria by inhibiting cell wall mucopeptide synthetase, resulting in the expansion and lysis of bacteria. According to the structural characteristics, β-lactams can be classified into two main types, including penicillins and cephalosporins, which are essential antibacterial chemotherapy drugs and are widely used in the treatment of bacterial infections because of their broad spectrum of activity, clinical effectiveness, and safety profile (Figure 1) [11,12]. Even if they are reputed to be products with low toxicity, the overuse and misuse of β-lactams may lead to residues in food and can contribute to public health problems such as allergic reactions, dysbiosis of the gut flora, and even carcinogenicity. The antibiotic-resistant bacteria that are pathogenic to humans may also occur, thereby presenting a significant challenge in infection treatment and control. Furthermore, β-lactam antibiotic residues in milk also interfere in fermentation processes used in cultured dairy products; e.g., yogurt. To improve food quality and ensure human health safety, the Commission Regulation (EU) No. 37/2010 has established maximum residue limits (MRL) of β-lactams in dairy products such as muscle, liver, kidney, fat, milk, or eggs [13].

Thus, various methods have been developed to determine the β-lactam residues that can be classified into three main categories: microbial inhibition, chromatographic techniques, and immunoassays [14,15,16,17,18]. Of these, the immunoassays are portable and cost-effective with high sensitivity and selectivity, and are capable of screening large numbers of samples [1,2]. However, it is quite difficult to produce antibodies that could achieve simultaneous identification of β-lactams due to the diversity of the molecular structure of these drugs [19,20]. Recently, research studies on β-lactam detection based on their receptors, termed as penicillin binding proteins (PBPs), which can recognize a variety β-lactams simultaneously, have been increasingly applied in β-lactam screening [21]. Some assays for the multi-residue analysis of β-lactams based on PBPs have been reported [22,23,24]. Noticeably, these methods are based on the competitive format, which is designed as reagent limited, always leading to poor sensitivity and with a narrow analytical range compared to non-competitive format.

In our previous study, we successfully expressed a recombinant PBP named PBP2x*, and a multi-residue competitive assay for detecting β-lactams was developed [24]. Earlier studies demonstrated that the spatial conformation of PBP2x* would change after the combination with β-lactams [25,26]. Therefore, in the present study, we explored the screening of anti-metatype antibodies for recognizing the complexes formed by PBP2x* and β-lactams, as well as the development of a sandwich immunoassay for β-lactams. Firstly, five β-lactams with high affinity to PBP2x* were selected to bind with PBP2x* to serve as immunogens. Subsequently, the monoclonal antibodies (mAb) and polyclonal antibodies (pAb) that could identify the PBP2x*–drug complexes were prepared, and the best pairing antibodies used as capture and detection antibodies in sandwich immunoassay were obtained, respectively. Eighteen β-lactams were combined with PBP2x* and then evaluated by the sandwich assay, and a real sandwich immunoassay for ceftriaxone (CTRX) detection with high sensitivity and wide detection range was successfully established (Figure 2). Furthermore, the insights into relationship between the structure of β-lactams and the conformational change of the PBP2x* were provided based on computational chemistry and molecular recognition study. To the authors’ knowledge, this work presents the first attempt to prepare anti-metatype antibodies for β-lactams based on PBP. More importantly, the study provides valuable references for further antibody screening and the development of more broad-spectrum non-competitive immunoassay for β-lactam detection.

## 2. Results and Discussion

### 2.1. Subsection

#### 2.1.1. Determination of the Binding Ability of β-Lactams to PBP2x*

The anti-metatype antibodies that recognize the complex formed by the receptor and ligand was developed by immunization [5]. The success of the immunization depends on the stability of the receptor–drug complex [9]. In the present study, we hypothesized that the complex formed by PBP2x* and β-lactams with high affinity to PBP2x* would achieve higher stability, because more non-covalent bonds typically tend to form between the two. In addition, the conformational change of PBP2x* induced by the combination of the β-lactams with high affinity may be greater due to the existence of more interaction forces, which stimulate the PBP2x* to some extent [27]. Therefore, a direct competitive ELISA was carried out to determine the affinity of 18 β-lactams to PBP2x*. As shown in Table 1, among the eight penicillins, PNG, AMP, AMX, AZL, and PIPC exhibited high affinity to PBP2x*, while OXA, CLX, and DCX showed relatively poor affinity. In the cephalosporin group, CPZ, CFN, CFZ, and CFM performed poorly: affinities with IC_50_ were greater than 10 ng/mL, which was almost 2–50-fold that of other drugs such as CTRX, CTX, and KF. (Table 2). Therefore, for the preparation of anti-metatype antibodies, five β-lactams (including PNG, AMP, AMX, CTX, and KF) with high affinity to PBP2x* were selected to form a complex with PBP2x*.

#### 2.1.2. Identification of Immunogens and Production of Anti-Metatype Antibodies

The quality of the immunogen is crucial for the antibody response. In this study, which differed from previous works [7,8,9], we used a receptor–drug complex as the immunogen, rather than a complex formed by an antibody and a drug. The central difference between the two is that an antibody usually identifies a specific target drug, while a receptor can recognize a class of drugs. Thus, five β-lactams (including PNG, AMP, AMX, CTX, and KF) with high affinity to PBP2x* were selected to form complexes with PBP2x*, which were then used as immunogens to immunize mice or rabbits to obtain antibodies against themselves. To ensure that all sites of PBP2x* were occupied by the drugs, the prepared complexes were identified by direct non-competitive ELISA. The results showed that there was almost no color development (OD values < 0.1, Appendix A) for five PBP2x*–drug complexes, indicating that the sites of PBP2x* had been fully occupied by drugs, which could be used as qualified immunogens for subsequent immunization.

The antisera of rabbits and mice immunized with five immunogens were screened against their homologous PBP2x*–drug complexes using indirect non-competitive ELISA, and the separate PBP2x* was also coated as control. Relatively high dilution of all 11 rabbit antisera were observed and collected after the fourth injection due to the death of one rabbit injected with immunogen PBP2x*–CTX. Among immunized mice, the antisera from PBP2x*–AMP yielded the highest titer, and thus was selected for cell fusion. After four subcloning processes using the limiting dilution method, a total of 31 hybridomas were obtained.

#### 2.1.3. Screening of Antibody Pairs

To screen the antibody pairs against the complex formed by PBP2x* and β-lactams, 11 pAbs and 31 mAbs were carried for sandwich immunoassays experiments, further leading to 321 antibody combinations for β-lactams to be tested. To improve the screening efficiency, seven β-lactams (PNG, AMX, AMP, OXA, CLX, CTX, and KF) were first chosen as the standard. Among the tested antibodies, the mAb 4H11 and pAb #5 derived from immunogen PBP2x*–CTX were selected as the capture and detection antibody, respectively, due to their better capacity to undergo sandwich immunoassays for OXA, CLX, and CTX. Thus, the selected pair of antibodies was employed to further evaluate the other 11 β-lactams. As shown in Figure 3a, aside from OXA, CLX, and CTX, the signal-to-noise (S/N) values of AZL, DCX, CTRX, CFT, CTX, CFP, CFM, and CPZ at a concentration of 1000 ng/mL were also greater than 1.0, indicating a trend of being detected by sandwich immunoassay. Among these, the CTRX yielded the best detection performance, reaching a high S/N value of 2.84. Therefore, a rapid and highly sensitive sandwich immunoassay for detection of CTRX was established. Under the optimized experimental conditions, standard solutions were diluted in milk samples without any previous dilution or centrifugation at concentrations of 0.1, 1, 10, 100, and 1000 ng/mL, and a standard curve of sandwich ELISA for CTRX detection was constructed with an LOD of 1.65 ng/mL and a working range of 1–1000 ng/mL (Figure 3b).

Theoretically, the anti-metatype antibodies recognize the receptor–ligand complex, but exhibit very low or no affinity for the receptor or the ligand alone. However, in the present study, the selected antibodies were observed to recognize the separate PBP2x* to some extent, resulting in high background values of 0.3–0.5. This may be related to the fact that the detection antibody used in the sandwich immunoassay was pAb, which contains antibodies that not only recognize the new epitopes created by the binding of PBP2x* and β-lactams, but also other epitopes of separate PBP2x*. Therefore, the use of pAb should be avoided whenever possible in further related studies. In addition, when applied to the 18 β-lactams, the sandwich immunoassay did not always detect drugs that exhibited high affinity to PBP2x*. Of the five immunized β-lactams (including PNG, AMP, AMX, CTX, and KF), only CTX showed an upward trend (S/N > 1) in the sandwich immunoassay when the CTX concentration was increased, while the other four high-affinity drugs failed to lead to sandwich immunoassays (S/N < 1) (Figure 3a). Conversely, the OXA, DCX, CLX, CFM, and CPZ, which possessed poor affinity with PBP2x*, seemed to be better recognized after combination with PBP2x* by sandwich immunoassay. The unexpected result drove us to further explore the binding mechanism between PBP2x* and β-lactams.

#### 2.1.4. Research on the Binding Mechanism of PBP2x* and β-Lactams

Initially, we assumed that the complexes consisting of high-affinity β-lactams and PBP2x* were more stable, and the greater conformation changes of PBP2x* might be induced, which was expected to obtain more effective anti-metatype antibodies and therefore more easily achieve sandwich immunoassay for β-lactam detection. However, the results were almost the opposite, and further exploration should be carried out. According to previous studies, the shape of the receptor binding pocket was originally influenced by the molecular structure and properties of the ligand, which in turn affected the subsequent production of the antibody against conformational change of receptor [28]. Therefore, on the basis of lowest energy conformations, we extracted physicochemical descriptors of 18 β-lactams including molecular volume/surface, Log P, etc., by computational chemistry. As shown in Appendix A, the β-lactams, such as DCX, PIPC, CPZ, CTRX, CFT, etc., which possessed lower energies, larger surface areas, and larger volumes, were observed to be more accessible to sandwich immunoassay. Therefore, the size and complexity of molecular structure may be a key factor affecting the conformational change of the receptor in the binding process.

To further explain the scientific base behind the new binding model we proposed, structural studies on 18 β-lactams in complex with crystal structure of PBP2x (PDB code: 2z2m) were conducted. As shown in Figure 4 and Appendix A, in the group of penicillins, the parent nucleus structure (6-aminopenicillanic acid) of eight drugs was deeply inserted into the binding cavity surrounded by the main residues ASN397, SER337, THR550, GLU334 PHE450, TRP374, and TYP561, while the C6 side chain pointed outward. We speculated that during the binding process, the parent nucleus moiety of penicillins firstly bound to PBP2x*, and in turn entered the binding cavity via the help of hydrogen bonding forces and hydrophobic forces, and the side chain moiety subsequently reached the cavity. In this context, despite that the interactions between the side chain moiety and the PBP2x* were less than the parent nucleus moiety, it still had an important effect on the formation of the binding cavity openings (Appendix A). The side chain of AZL, PIPC, OXA, DCX, and CLX is a double-ring structure, which is more complex and more rigid than the single-ring structure of PNG, AMX, and AMP. Coincidentally, the drugs with a more complex side chain had a greater probability of being recognized by the antibody after binding to the receptor, as shown in Figure 3a. Among these, the benzo isoxazole structure of OXA, DCX, and CLX showed rotations of almost 90° in the binding cavity, which indicated low fitness with PBP2x*, thus belonging to the low-affinity ligand. However, such deflections induced by rigid structure of drugs may easily cause structural changes of the receptor in the binding cavity, and thereby more likely to be identified by anti-metatype antibodies. Similar results were observed in the cephalosporin group: the cephalosporins with more complex structures in the C3 and C7 side chains yielded better detection efficacy by sandwich immunoassay after binding to PBP2x*, such as CTRX, CFT, CFP, CPZ, and CFM. As shown in Figure 5, the C3 side chain of cephalosporins protrudes deep into the cavity, whereas the C7 side chain moiety of the hapten is partially exposed to solvent, which indicated the criticality of the C3 side chain in inducing a conformational change of PBP2x*. Of these, the CTRX exhibited the best detection effect by sandwich immunoassay, which most likely was facilitated by the more complex C3 and C7 side chains that formed multiple hydrogen bonds with PBP2x* (Appendix A). In total, the sandwich immunoassay results, together with molecular recognition study, demonstrated that the complexity of the molecular structure of the drug affected the conformational changes of the receptor after binding, which in turn affected the subsequent sandwich detection. Moreover, in the process of screening optimal antibody pairs for establishing the sandwich immunoassay, the sandwich detection could be achieved only when the pAb derived from PBP2x*–CTX was used as the detection antibody, coupled with mAb 4H11 from PBP2x*–AMP as the capture antibody. In comparison to PNG, AMP, AMX, and KF, the CTX, with a more complex and rigid structure, exhibited the stronger induction ability to induce conformational changes of the receptor, and thereby promoted the generation of anti-metatype antibodies. Therefore, in follow-up association studies, the β-lactams with a larger size and more complex structures should be selected for binding to the receptor and then used as the immunogen to screen anti-metatype antibodies, and hence achieve broad-spectrum reorganization for β-lactams using a sandwich immunoassay.

## 3. Materials and Methods

### 3.1. Materials

Penicillin G (PNG), ampicillin (AMP), amoxicillin (AMX), azlocillin (AZL), oxacillin (OXA), piperacillin (PIPC), cefalothin (KF), cefazolin (CFZ), ceftiofur (CFT), cefepime (CFP), and cefixime (CFM) were purchased from the National Institutes for Food and Drug Control (Beijing, China). Cloxacillin (CLX), dicloxacillin (DCX), cefoperazone (CPZ) and cefotaxime (CTX) were purchased from Dr. Ehrenstorfer (Augsburg, Germany). Cefalonium (CFN), CTRX, cefapirin (CFAP), horseradish peroxidase (HRP), poly (ethylene glycol) (PEG) 1500, hypoxanthine aminopterin thymidine (HAT), incomplete Freund’s adjuvant (IFA), and complete Freund’s adjuvant (CFA) were purchased from Sigma-Aldrich (St. Louis, MO, USA). Fetal calf serum and Dulbecco’s Modified Eagle’s Medium (DMEM) were obtained from Gibco BRL (Carlsbad, CA, USA). The 3,3′,5,5′–tetramethylbenzidine (TMB) and goat anti-mouse IgG were acquired from Jackson ImmunoResearch (West Grove, PA, USA). The HRP conjugation labeling kit was from Abcam (Toronto, Canada). Reagent grade solvents and salts were supplied by Beijing Chemical Reagent Co. (Beijing, China). The buffer solutions are listed in the Appendix A.

### 3.2. Determination of Binding Affinity of PBP2x* with 18 β-Lactams

The PBP2x* (100 μL/well) was diluted to an appropriate concentration and coated on microtiter plates, and incubated overnight at 4 °C. After the solution was poured, the plates were washed 3 times with washing buffer and patted dry. Then, 150 μL of blocking solution was added to each well, followed by 1 h of incubation at 37 °C. The β-lactam standards (50 μL), which were diluted to a series of concentrations, were added into each well, then 50 μL of the enzyme marker prepared in our previous study [24] was added, and incubated at 37 °C for 30 min. After the washing procedure, the color was developed by adding tetramethylbenzidine (TMB) substrate. Finally, 2 M H_2_SO_4_ (50 μL/well) was used to stop the enzymatic reaction, and the optical density (OD) values of 450 nm was measured.

### 3.3. Preparation and Identification of PBP2x*—Drug Complexes

The purified PBP2x* (10.0 mg) was dissolved in 5 mL PBS, followed by adding 10-fold molar excess of β-lactams (PNG, AMP, AMX, CTX, and KF). The reaction mixture was incubated for 24 h at 4 °C and dialyzed against PBS for 3 days. To ensure that the active sites of the PBP2x* (receptor) were completely occupied by the β-lactams (ligand), they were identified by a direct enzyme-linked immunosorbent assay (ELISA). Briefly, the microplates were first coated with the receptor–ligand complex (100 μL/well), which was diluted with CB buffer and then incubated at 4 °C overnight. The coating solution was then discarded and washed by washing buffer. Blocking buffer was then added to the plates (150 μL/well) and incubated for 2 h at 37 °C. After patting to dry, 100 μL diluted PNG-HRP, which was prepared in our previous study [24], was then added and incubated for 30 min at 37 °C. TMB substrate (100 μL/well) was added and incubated for 15 min at 37 °C, and then 2 M H_2_SO_4_ (50 μL/well) was used for reaction stopping, and the OD values at 450 nm were measured.

### 3.4. Preparation of Anti-Metatype Antibody

#### 3.4.1. Rabbit pAbs

Female New Zealand white rabbits were used to produce pAbs. Five complexes (including PBP2x*–PNG, PBP2x*–AMP, PBP2x*–AMX, PBP2x*–CTX, and PBP2x*–KF) and individual PBP2x* were used as immunogens to immunize two rabbits. The immunization procedure was described in our previous studies [29]. The titers of antisera were monitored by non-competitive ELISA. Briefly, the ELISA plates were coated with the corresponding PBP2x*–drug complex (100 μL/well) and incubated at 4 °C for overnight. After blocking, 100 μL of diluted serum samples was added and incubated at 37 °C for 1 h. The HRP-labeled goat anti-rabbit IgG (100 μL /well) was added after the washing step, and the plates were incubated for 1 h at 37 °C. Then, plates were washed three times, followed by a color development. The OD values of each well were measured at 450 nm. A positive antibody response was defined as an OD value higher than 2.1 times the mean of OD values of the serum samples from rabbit immunized with PBP2x*.

#### 3.4.2. Mouse mAbs

Five eight-week-old female Balb/c mice were first immunized with 100 µg of each immunogen in 0.1 mL of PBS and 0.1 mL of CFA. For the second immunization, the mice were boosted with the mixture of immunogen (100 µg) and IFA. After the fourth immunization, the mouse with the highest dilution was sacrificed, and the splenocytes were extracted and fused with SP2/0 myeloma cells. The hybridoma cells were screened for antibody production using non-competitive ELISA as described above, followed by subcloning based on the limiting dilution method. The clones with the high inhibition were cloned third, prior to ascites production. The mAb was purified by saturated ammonium sulfate precipitation from ascites and stored at −20 °C.

### 3.5. Screening for Pairing Antibodies

Microtiter plates were coated with 100 μL anti-mouse IgG antibody in CB buffer and incubated at 4 °C overnight. The plates were subsequently washed and blocked with 150 μL of 2% skim milk. Then, 100 μL of diluted mAb was added to the wells and incubated for 1 h at 37 °C. After washing, 100 μL individual PBP2x* or receptor–ligand complexes of 18 β-lactams with PBP2x* that were diluted to 1 μg/mL was added. The incubation was carried out at 37 °C for 1 h, followed by another washing procedure. Then, 100 μL diluted HRP-labeled pAb was added to the wells and incubated for 1 h at 37 °C. The TMB substrate solution was added after washing, followed by the addition of 2 M H_2_SO_4_ to stop the reaction, and finally the OD values were measured at 450 nm.

#### Sandwich Immunoassay for CTRX

Microtiter plates were coated with anti-mouse IgG antibody first, and then the mAb diluted to the appropriate concentration was added and incubated at 37 °C for 1 h. At the same time, appropriately diluted PBP2x* was incubated with different concentrations of CTRX (0.1, 1, 10, 100, or 1000 ng/mL) at 37 °C for 1 h. Then, the wells coated with mAb were washed, and the prepared PBP2x*–drug complexes were added. The incubation was carried out at 37 °C for 1 h, followed by the addition of HRP-labeled pAb. After 1 h of incubation, the color was developed by adding 100 μL of the TMB substrate solution. The solution was incubated for 15 min at 37 °C before the enzymatic reaction was stopped by adding 50 μL of 2 M H_2_SO_4_. The OD values of each well was measured at 450 nm.

### 3.6. Extraction of Molecular Descriptors

Three-dimensional (3D) structures of β-lactam molecules were built in GaussView 5.0 software (Gaussian, Wallingford, CT, USA). Then, all molecules were optimized by density functional theory (DFT) calculations at the TVZP functional level with an M06-2X basis set using the Gaussian 09 software (Gaussian, Wallingford, CT, USA) [28,30]. The descriptors of β-lactams, including the energy (E) and dipole moment (μ), were extracted from the Guassian output file. The hydrophobic constant (Log P) was obtained using ChemDraw (PerkinElmer, Waltham, MA, USA). The polar surface area (PSA), molecule volume (Vm), molecular polarity index (MPI), and surface area (SA) were extracted using Multiwfn 3.7 (dev) code [31].

### 3.7. Molecular Docking between the β-Lactams and PBP2x*

The bindings between 18 β-lactams and the PBP2x* (PDB code: 2z2m) were performed by CDOCKER (Discovery Studio 17.1, Dassault Systèmes, BIOVIA Corp., San Diego, CA, USA). After docking ligands into the top-ranked cavity of the antibody, the best binding pose with the highest score for each ligand was selected.

## 4. Conclusions

In the current study, we presented the first attempt to prepare anti-metatype antibodies for the complex formed by PBP2x* and β-lactams, and developed a sandwich immunoassay for β-lactam detection. It was found that the affinity between PBP2x* and β-lactams did not seem to be a key factor in the capacity of antibody screening and sandwich immunoassay development, but was closely related to the size and complexity of the molecular structure of drugs. Therefore, in the follow-up study, the complex of PBP2x* and β-lactams with a relatively complex molecular structure should be selected as the immunogen to obtain paired antibodies that can be used for sandwich detection, which is expected to significantly improve the detection performance. This study will provide practical experience and a theoretical basis for subsequent screening of more broad-spectrum and sensitive antibodies against β-lactams.

## Figures and Tables

**Figure 1 molecules-26-05569-f001:**
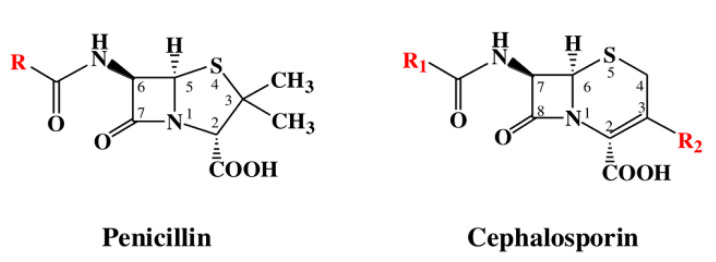
The structures of penicillin and cephalosporin. The β-lactam scaffolds are shown in black, and the different side chain structures of β-lactam are shown as R in red.

**Figure 2 molecules-26-05569-f002:**
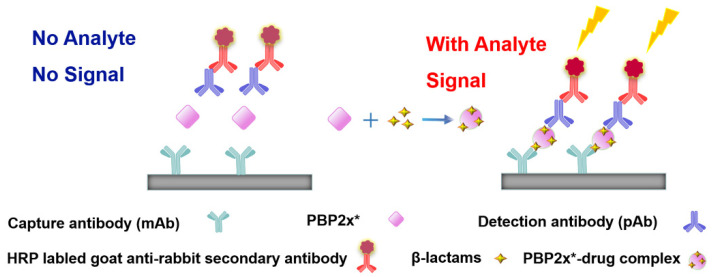
Schematic representation of sandwich immunoassay for small-molecule detection based on PBP2x*.

**Figure 3 molecules-26-05569-f003:**
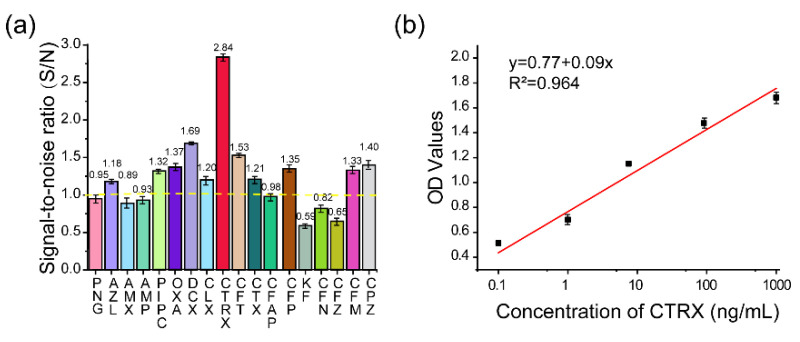
(**a**) The S/N of the sandwich immunoassay for detection of 18 β-lactams (*n* = 3). S/N > 1 indicates positive detection by the sandwich immunoassay, while S/N ≤ 1 indicates negative detection. (**b**) Standard curve of the sandwich immunoassay for CTRX detection (*n* = 3).

**Figure 4 molecules-26-05569-f004:**
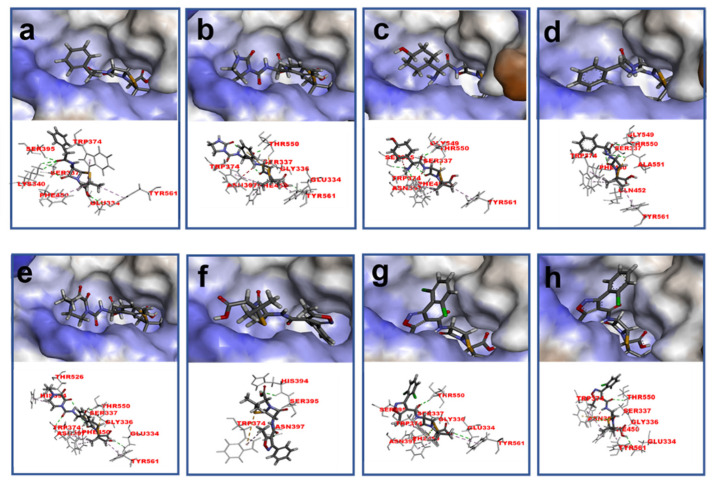
Schematic representation of the interactions between PBP2x* (PDB code: 2z2m) and (**a**) PNG; (**b**) AZL; (**c**) AMX; (**d**) AMP; (**e**) PIPC; (**f**) OXA; (**g**) DCX; and (**h**) CLX. Hydrogen bonds are shown as green dashed lines, and hydrophobic interactions are displayed as purple dashes.

**Figure 5 molecules-26-05569-f005:**
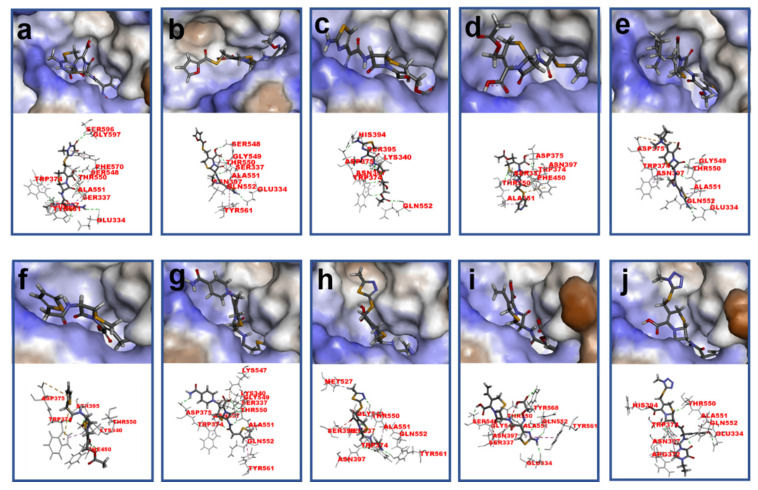
Schematic representation of the interactions between PBP2x* (PDB code: 2z2m) and (**a**) CTRX; (**b**) CFT; (**c**) CTX; (**d**) CFAP; (**e**) CFP; (**f**) KF; (**g**) CFN; (**h**) CFZ; (**i**) CFM; and (**j**) CPZ. Hydrogen bonds are shown as green dashed lines, and hydrophobic interactions are displayed as purple dashes.

**Table 1 molecules-26-05569-t001:** The affinity of penicillins to PBP2x*.

Penicillins	Structures of R	IC_50_ (ng/mL)
Penicillin G (PNG)	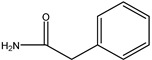	3.56
Azlocillin (AZL)	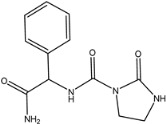	6.67
Amoxicillin (AMX)	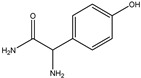	6.84
Ampicillin (AMP)	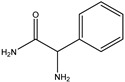	7.78
Piperacillin (PIPC)	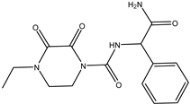	7.78
Oxacillin (OXA)	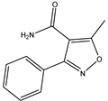	10.08
Dicloxacillin (DCX)	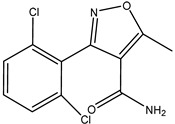	15.31
Cloxacillin (CLX)	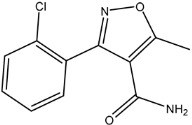	75.23

**Table 2 molecules-26-05569-t002:** The affinity of cephalosporins to PBP2x*.

Cephalosporins	Structures of R1	Structures of R2	IC_50_ (ng/mL)
Cefatriaxone (CTRX)	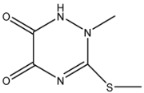	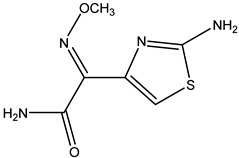	2.35
Ceftiofur (CFT)	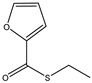	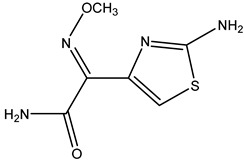	2.63
Cefotaxime (CTX)	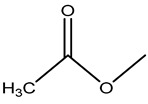	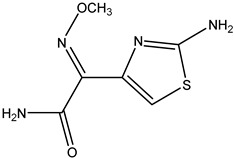	4.75
Cefapirin (CFAP)	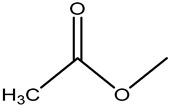	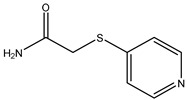	6.05
Cefepime (CFP)	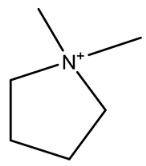	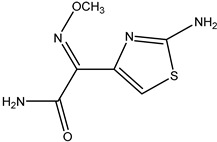	7.32
Cefalothin (KF)	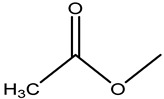	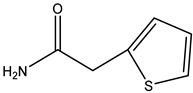	7.89
Cefalonium (CFN)	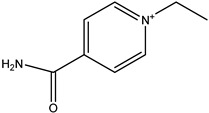	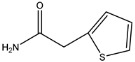	15.30
Cefazolin (CFZ)	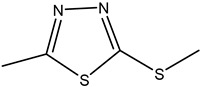	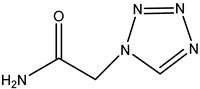	24.36
Cefixime (CFM)	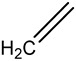	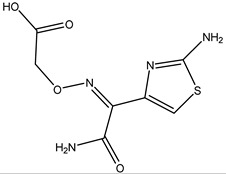	29.33
Cefoperazone (CPZ)	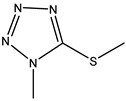	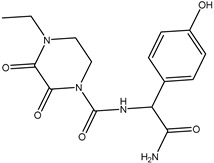	100.22

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
