# Peer review of "Anti-Metatype Antibody Screening, Sandwich Immunoassay Development, and Structural Insights for β-Lactams Based on Penicillin Binding Protein"

_molecules, 2021, doi:10.3390/molecules26185569_

Round 1

Reviewer 1 Report

The article “Structural insights into the conformational changes of receptor induced by small-molecule ligand: A case study on β-lactams “ by Yuchen B. et al. presents the development of a novel strategy for sandwich immunoassay of small molecules, such as B-lactams.

Although b-lactams are classic antibodies with low toxicity, an excess concentration in food can lead to allergic reactions. Therefore, the development of a method to detect this type of product in food is of utmost importance.

The authors develop a method based on a sandwich immunoassay to detect B-lactams in PBS samples. 

Overall the article is well presented, and the results are consistent. Nevertheless, I have some minor revisions that are necessary to be addressed.

  1. Although in the introduction, the authors mention the relevance of detecting b-lactams, the paragraph needs to be better explained. (line 51-62, precisely 56-62)
  2. All the studies were perfumed using PBS buffer and the calibration curve to detect CTRX (Figure 3B). It is relevant to analyse the impact of complex samples (such as food samples) in this assay. 
  3. In section 2.1.3, Please include the linear regression of Figure 3B and the correlation R2 value. 

Author Response

Thank you for the general comment. We have carefully revised the manuscript according to these comments. Detailed responses to review comments are attached below.

  1. Although in the introduction, the authors mention the relevance of detecting b-lactams, the paragraph needs to be better explained. (line 51-62, precisely 56-62).

Reply: Thank you very much for your comments.

  We have revised the line 51-62 in Introduction and marked red in revised manuscript.

β-lactams refer to a class of antibiotics with β-lactam ring in their chemical structure, which is a kind of drugs that destroy the cell wall of bacteria by inhibiting cell wall mucopeptide synthetase, resulting in the expansion and lysis of bacteria. According to the structural characteristics, β-lactam can be classified into two main types, including penicillins and cephalosporins, which are an essential part of antibacterial chemotherapy drugs and widely used in the treatment of bacterial infections because of their broad spectrum of activity, clinical effectiveness and safety profile (Figure 1) [11-12]. Even if they are reputed to be products of low toxicity, the overuse and misuse of β-lactams may lead to residues in food and can contribute to public health problems such as allergic reactions, dysbiosis of the gut flora, and even carcinogenicity. The antibiotic-resistant pathogenic bacteria to human may be also occurred, thereby presenting a significant challenge in infection treatment and control. Furthermore, β-lactam antibiotic residues in milk also interfere in fermentation processes used in cultured dairy products, e.g. yogurt. To improve the food quality and ensure human health safety, the Commission Regulation (EU) No 37/2010 has established maximum residue limits (MRL) of β-lactams in dairy products such as muscle, liver, kidney, fat, milk or eggs [13].

  1. All the studies were perfumed using PBS buffer and the calibration curve to detect CTRX (Figure 3B). It is relevant to analyze the impact of complex samples (such as Antibiotics are widely used in veterinary medicine for the treatment or prophylaxis of bacterial infections, as well as for growth promotion. In dairy production, antibiotics fromβ-lactam class are the most used drugs to treat mastitis. food samples) in this assay.

Reply: Thank you very much for your comments.

Currently, β-lactam antibiotics are still frequently used in veterinary practice for the prevention and treatment of bacterial infections, especially in mastitis therapy of dairy cow [1-2]. The b-lactam antibiotics presented in milk may result in a potential hazard to consumers and technological problems in the dairy industry. Therefore, we detected CTRX in milk samples after simple pretreatment (centrifugation to remove fat), which exhibited a similar LOD value (1.65 ng/mL) with that in PBS (1.62 ng/mL). We have added the data to the revised manuscript and marked as red.

[1] Zhang J, Wang Z, Wen K, Liang X, Shen J. Penicillin-binding protein 3 of streptococcus pneumoniae and its application in screening of β-lactams in milk. Anal Biochem. 2013; 442: 158–165.

[2] Pitts CR, Lectka T. Chemical synthesis of beta-lactams: asymmetric catalysis and other recent advances. Chem Rev. 2014; 114: 7930–7953.

  1. In section 2.1.3, Please include the linear regression of Figure 3B and the correlation R2 value.

Reply: Thank you very much for your comments.

We have added the linear regression and R2 value in revised Figure3B.

y=0.77+0.09x

R2=0.964

Reviewer 2 Report

Dear Editor,

The manuscript entitled “Structural insights into the conformational changes of receptor induced by small-molecule ligand: A case study on β-lactams” by Yuchen Bai et al. presents a novel strategy for sandwich immunoassay of small molecules design and development. The authors prepare anti-metatype antibodies for β-lactams based on receptor PBP2x*, evaluate the antibody pairs performance and provide a structural study the conformational changes of β-lactams and PBP2x* interactions based on computational chemistry.

In my opinion, the manuscripts’ objective and findings are very interesting, the study is well designed and adequately explained and the manuscript is satisfyingly written. Therefore I think it should be accepted for publication after minor revisions. My detailed comments for the authors to consider are provided below:

  1. I would recommend to change the manuscript title since developing immunoassay based on anti-metatype antibodies is the essence of the study and the structural insights are only part of the presented work. The present title can be considered misleading.
  2. The abstract needs to be re-written: it is a little bit complicated; I understood it only after I had read the whole manuscript which does not serve the purpose of an abstract. Please write more comprehensively the study steps and findings.
  3. In page 3, lines 109-114, please re-phrase to clarify the findings.
  4. In page 3, lines 124-127, please provide the ELISA values as supplementary material.
  5. In figure 3(a), the graph should contain the error bars.
  6. In page 9, line 270, what the appropriate concentration and how it is determined?
  7. In page 9, line 275, which is the added enzyme marker?
  8. In page 10, line 312, briefly explain the procedure and the ELISAs.
  9. In page 10, line 323, how were the pAbs labelled with HRP? You need a brief description and/ or reference
  10. In page 10, line 328, what the appropriate concentration and how it is determined?

Author Response

Thank you for the general comment. We have carefully revised the manuscript according to these comments. Detailed responses to review comments are attached below.

  1. I would recommend to change the manuscript title since developing immunoassay based on anti-metatype antibodies is the essence of the study and the structural insights are only part of the presented work. The present title can be considered misleading.

Reply: Thank you very much for your comments.

We have revised the title of manuscript as “The anti-metatype antibody screening, sandwich immunoassay development, and structural insights for β-lactams based on penicillin binding protein” to present the work better.

  1. The abstract needs to be re-written: it is a little bit complicated; I understood it only after I had read the whole manuscript which does not serve the purpose of an abstract. Please write more comprehensively the study steps and findings.

Reply: Thank you very much for your comments.

We have re-written the abstract to be more comprehensively and fulfill the role of an abstract

Theoretically, sandwich immunoassay is more sensitive and has a wider working range than that of competitive format. However, it has been thought that small molecules cannot be detected by sandwich format owing to their limited size. In the present study, we proposed a novel strategy for achieving sandwich immunoassay of β-lactams with low molecular weights. Firstly, five β-lactam antibiotics were selected to bind with penicillin binding protein (PBP)2x* to form complexes, respectively. Then, monoclonal and polyclonal antibodies against PBP2x*-β-lactams complexes were produced by animal immunization. Subsequently, the optimal pairing antibodies were utilized to establish sandwich immunoassay for eighteen PBP2x*-β-lactams complexes detection. Among them, ceftriaxone can be detected as low as 1.26 ng/mL with working range of 1-1000 ng/mL in milk. To reveal the detection mechanism, computational chemistry and molecular recognition study were carried. The results shown that β-lactams with large size and complex structures maybe conducive to induce conformational changes of PBP2x*, and then exhibit greater possibility of being detected by sandwich immunoassay after combination with PBP2x*. This study provides insights for subsequent investigations of anti-metatype antibody screening and sandwich immunoassay establishment for small molecule detection.

  1. In page 3, lines 109-114, please re-phrase to clarify the findings.

Reply: Thank you very much for your comments.

We have revised the results to be more clearly.

As shown in Table 1, among the 8 penicillins, PNG, AMP, AMX, AZL, and PIPC exhibited high affinity to PBP2x*, while OXA, CLX, and DCX showed relatively poor affinity. Besides, in cephalosporins group, CPZ, CFN, CFZ, and CFM performed poor affinities with IC50 were greater than 10 ng/mL, which was almost 2-50-fold than that of other drugs such as CTRX, CTX, and KF. (Table 2). Therefore, for the preparation of anti-metatype antibodies, 5 β-lactams (including PNG, AMP, AMX, CTX, and KF) with high affinity to PBP2x* were selected to form complex with PBP2x*.

  1. In page 3, lines 124-127, please provide the ELISA values as supplementary material.

Reply: Thank you very much for your comments.

The ELISA values of detected 5 PBP2x*-drug complexes have been added in supplementary material.

  1. In figure 3(a), the graph should contain the error bars.

Reply: Thank you very much for your comments.

The error bars have been added to the Figure3A.

  1. In page 9, line 270, what the appropriate concentration and how it is determined?

 Reply: Thank you very much for your comments.

The dilution of the PBP2x was 1/300, and we determined the appropriate dilution based on the checkerboard assays to get the lowest IC50. Briefly, the PBP2x was diluted to 1/100, 1/300, 1/900, 1/2700, 1/8100 and coated on microtiter plates. Then, the analyte and enzyme marker were added and the competitive reaction was occurred for binding to the PBP2x*.

  1. In page 9, line 275, which is the added enzyme marker?

Reply: Thank you very much for your comments.

The enzyme marker is the conjugate composed of analyte and HRP which prepared in our previous study [1] and added to the revised manuscript.

[1] Zeng, K., Zhang, J., Wang, Y., Wang, Z.H., Zhang, S., Wu, C., & Shen, J. Development of a rapid multi-residue assay for detecting beta-lactams using penicillin binding protein 2x*, Biomed Environ Sci: BES. 2013; 26:100–109.

  1. In page 10, line 312, briefly explain the procedure and the ELISAs.

 Reply: Thank you very much for your comments.

Briefly, each well of microtiter plates was first coated with 100μL of PBP2x-amp complex in carbonate buffer and then incubated at 4°C overnight. After blocking, 100 μL of antibodies secreted by hybridoma cells were added to the wells, and incubation for 30 min at 37°C was carried out. After washing two times, 100μL of diluted HRP-labeled goat anti-mouse IgG was then added followed by incubation for 30 min at 37°C. The TMB substrate was added and incubated for 15 min at 37°C after further washing. Then the chromogenic reaction was inhibited using 2 M H2SO4(50μL/well), and the optical density (OD) values of 450 nm were measured.

  1. In page 10, line 323, how were the pAbs labelled with HRP? You need a brief description and/ or reference

Reply: Thank you very much for your comments.

The pAbs were lebelled with HRP according to the description of HRP Conjugation Kit/HRP Labeling Kit (Abcam, ab102890)., which has been added in revised manuscript.

  1. In page 10, line 328, what the appropriate concentration and how it is determined?

Reply: Thank you very much for your comments.

The optimal concertation of mAb was 1/1000, which determined by checkerboard assay as described above to get lowest LOD value.